# The Possible Mechanism of Amyloid Transformation Based on the Geometrical Parameters of Early-Stage Intermediate in Silico Model for Protein Folding

**DOI:** 10.3390/ijms23169502

**Published:** 2022-08-22

**Authors:** Irena Roterman, Katarzyna Stapor, Dawid Dułak, Leszek Konieczny

**Affiliations:** 1Department of Bioinformatics and Telemedicine, Jagiellonian University—Medical College, Medyczna 7, 30-688 Kraków, Poland; 2Department of Applied Informatics, Faculty of Automatic, Electronics and Computer Science, Silesian University of Technology, Akademicka 16, 44-100 Gliwice, Poland; 3ABB Business Services Sp. z o.o. ul., Żegańska 1, 04-713 Warszawa, Poland; 4Chair of Medical Biochemistry—Jagiellonian University—Medical College, Kopernika 7, 31-034 Kraków, Poland

**Keywords:** amyloid, secondary structure, α-helix unfolding, β-structure, left-handed α-helix, hydrogen bonds, transthyretin, protein folding, early stage of folding, misfolding

## Abstract

The specificity of the available experimentally determined structures of amyloid forms is expressed primarily by the two- and not three-dimensional forms of a single polypeptide chain. Such a flat structure is possible due to the β structure, which occurs predominantly. The stabilization of the fibril in this structure is achieved due to the presence of the numerous hydrogen bonds between the adjacent chains. Together with the different forms of twists created by the single R- or L-handed α-helices, they form the hydrogen bond network. The specificity of the arrangement of these hydrogen bonds lies in their joint orientation in a system perpendicular to the plane formed by the chain and parallel to the fibril axis. The present work proposes the possible mechanism for obtaining such a structure based on the geometric characterization of the polypeptide chain constituting the basis of our early intermediate model for protein folding introduced formerly. This model, being the conformational subspace of Ramachandran plot (the ellipse path), was developed on the basis of the backbone conformation, with the side-chain interactions excluded. Our proposal is also based on the results from molecular dynamics available in the literature leading to the unfolding of α-helical sections, resulting in the β-structural forms. Both techniques used provide a similar suggestion in a search for a mechanism of conformational changes leading to a formation of the amyloid form. The potential mechanism of amyloid transformation is presented here using the fragment of the transthyretin as well as amyloid Aβ.

## 1. Introduction

The problem defined as misfolding—a phenomenon associated mainly with the appearance of the fibrillary form of amyloid proteins—is closely related to the problem of protein folding, which, despite significant progress, still cannot be treated as solved [1,2]. Significant progress in this field has been achieved by using machine learning techniques, especially deep learning [3,4,5], the most spectacular example of which is the AlphaFold project [6,7,8,9]. Progress in predicting protein structures based on the given sequence is being tracked thanks to the CASP project [10,11,12]. Despite the success achieved by the AlphaFold project in the form of correctly predicted structures for a given amino acid sequence, the search for the mechanism of the folding process that answers the question “*Why do the proteins fold the way they do?”* is still valid. The ab initio methods focus on the search for such a form of the force field, the use of which would reflect the mechanism of obtaining a 3D structure ensuring the performance of an appropriate biological function [13,14,15,16,17,18,19]. The use of the comparative modeling methods, despite the correct results obtained with this technique, also leaves the question about the folding mechanism unanswered [20,21,22,23]. The combination of techniques used by leading protein folding teams in WeFold in a manner comparable to AlphaFold did not answer the question about the folding mechanism [24,25].

It is possible that the analysis of the misfolding mechanism will prove to play an important role in solving the problem of protein folding. Apart from techniques based on numerical methods, the study of amyloids would not be possible without continuously developing experimental techniques [26,27,28,29,30,31].

This work, based on the analysis of the conformation of amino acids present in amyloid chains, proposes the mechanism of amyloid transformation with the use of our already developed and published two-stage model of protein folding, in particular the early stage of this process [32,33], described shortly in the Materials and Methods section as well as in the Appendix A–Appendix A.

The analysis of amyloid structure presented here focuses on defining:The specificity of the structure of the amyloid form of the polypeptide chain—the fibril component;The mechanism of structural transformation leading from the native form of the amyloid (transthyretin used here as an example) chain to its amyloid form using the previously proposed two-step model of protein folding. The analysis presented now mainly uses the early-stage model.

In the present study, the structure of transthyretin in its native form was analyzed as a reference structure (1DVQ—[34]) against two variants of this protein: aggressively undergoing amyloid transformation (1G1O—[35]) and the resistance to this transformation (1GKO—[36]). The distinguishing features between these two forms have been identified experimentally [35,36]. The structure (the conformation expressed by the Phi and Psi angles) of amyloid form of this protein (6SDZ—[37]) is the main point of the current analysis. 

The foundation of the analysis presented here is based on the following two approaches. 

Our model of early-stage protein folding [32,33], described shortly in the Materials and Methods section as well as in Appendix A–Appendix A. According to this model, it is assumed that amyloid transformation requires the unfolding treated as return to its early-stage form of folding.Simulation of α-helix unfolding by molecular dynamics performed and presented by Daggett and Levitt [38,39], described shortly below. This simulation is treated as an independent suggestion for the path expressing the R-α-helix unfolding process.

The molecular dynamics simulation of R-α-helix unfolding performed by Daggett and Levitt and presented in [38,39] revealed two paths depending on the status of the α-helix fragment under consideration. These two forms of α-helix were as follows: independent 13-alanine α-helix and α-helical fragment of the BPTI chain (Figure 1). In both cases, the β structure has been reached. 

The α-helical section constituting a fragment of BPTI protein [38,39,40] (Figure 1A) is subjected to the specific constraints (disulfide bonds limiting the freedom of the movement of terminal positions in the α-helical section and the fact that the α-helix is also bounded to the other parts of the protein’s chain) undergoes a structural transformation by reducing the radius of curvature (compression of the α-helix) by passing through conformation helix 3_10_ and helix 2_7_ (line G1, G2 and L2—Figure 2) [41,42]. 

A section of the free α-helix in the form of an isolated 13-alanine polypeptide with the starting structure of the right-handed α-helix (Figure 2), not subjected to any external constraints, unfolds by unfolding the α-helix, which is associated with a gradual increase in the radius of curvature (line L1—Figure 2). The discussion of radius change is explained in Materials and Methods,

In both cases, a β-structural form is obtained. Further increasing the temperature in the simulation leads to left-handed α-helix structures (lines L2 and G2). 

Based on the observed transition of the α-helical conformation into the β-structural and L-handed α-helix, it is assumed that the proposed two paths can be used as potential pathways leading to the formation of the β-dominated structure. Since the α-helix unfolding in transthyretin (as well as in any other protein undergoing the amyloid transformation) is part of a longer chain—the free movement is limited by the constraints—the path L2/G2 is discussed in this paper together with the ellipse path identified using the early-stage model for protein folding (Materials and Methods). 

Figure 3A–C present some elements from our early-stage model: the distribution of geometrical parameters R, V (i.e., the radius of curvature R and the angle V between the consecutive peptide bond planes) and the zones corresponding to the structural codes with the elliptical path. The paths representing structural changes in Figure 2 with respect to the parameters R, V are shown in Figure 3D,E. The stepwise increase in radius R along the paths L1 and G1 and decrease along the paths L2 and G2 can be observed (before the radical increase entering into β-structural area). The two aforementioned forms of unfolding of the α-helix are presented as paths in Figure 3D,E, where the conformations of helix 3_10_ and 2_7_ are marked, showing the reduction in the radius R in the case of paths G2 and L2 (Figure 3D,E) [32,33,41,42]. It can be easily seen that the structures in F zone represent forms of lower radius of curvature than in E zone. 

Based on the coherent results from the above-described molecular dynamic simulation with those based on our early-stage in silico model of protein folding (presented in Materials and Methods), we have posed our hypothesis on the possible form of the amyloid transformation mechanism.

The currently proposed mechanism of amyloid transformation uses the presented models. The analyzed conformational changes leading to the conversion of the native form into the amyloid form for a fragment of the chain were modeled in the present work according to the paths discussed. The background for zone definition is available in Appendix A–Appendix A.

## 2. Results

### 2.1. Analysis of the Specificity of Amyloid Structure Based on Transthyretin (PDB ID—6SDZ) 

Before introducing the presentation of our proposed method for conformational changes leading to amyloid formation, the specificity of the target structure will be given based on the transthyretin chain available in PDB (PDB ID—6SDZ)—Figure 4. The flat structure of the polypeptide chains present in amyloid fibrils is the simple consequence of a dominant presence of β-strands in the form of β-sheet structure, which is characterized by a highly ordered system of C=O and N-H groups forming hydrogen bonds. A similar system of ordered hydrogen bonds is present in α-helix structure. Both mentioned secondary structures involve the maximum number of C=O… H-N arrangements to build a hydrogen bond network, where each peptide plane uses both the C=O and H-N groups to form a hydrogen bond. In the case of the β-sheet, the hydrogen bonds remain anti-parallel, while in the α-helical form (for both the right- and left-handed α-helix), the system is parallel (Figure 3A,B). The β-structure system favors the construction of a flat rectilinear structure. It turns out, however, that the flat linear form of the chain is obtained by the selected combinations of the Phi and Psi angles introducing an anti-parallel system of hydrogen bonds. This form is obtained for the angles satisfying relation Psi = −Phi (in part of the Ramachandran map for Phi < 0 and Psi > 0). Such arrangement of torsion angles corresponds to a structure characterized by a V-angle close to 180 deg and the maximum radius of chain curvature (Figure 4A,B). Any other arrangement of Phi and Psi angles introduces a decrease in the radius of curvature, which results in the appearance of a structure of 3D form—the spatial one—while in amyloid, the structure is limited to 2D construction of an individual chain—a flat structure. 

The relation Psi = −Phi does not denote any change. It represents the specific set of Phi and Psi angles localized on the Ramachandran map as shown in Figure 5.

In the case of an α-helix with a parallel system of hydrogen bonds, a three-dimensional structure appears (successive turns of the α-helix). Full involvement of the C=O and N-H groups of peptide bonds in the formation of a hydrogen bond network maintaining a flat structure is possible for the β structure under restricted conditions. The single individual residue with R- or L-α-helical conformation is able to maintain a parallel system of hydrogen bonds together with relatively long fragments with β-structural form. The orientation of hydrogen bonds in β structure—the antiparallel one—is maintained by R- or L-handed α-helical conformation, reorienting only the dipoles of the hydrogen bonds in the parallel form. The final form does not disrupt the flat organization of the chain structure, while introducing the turns and specific forms of loops. 

Table 1 presents the proposed classification of the identified, specific structural forms (turns and forms different from β structure in amyloids) leading to the maintenance of a planar structure of chains present in amyloid fibrils (Figure 4B).

The flat structure of a single transthyretin chain in amyloid fibril is shown in Figure 4A with distinguished fragments representing forms other than β-structural forms. Through an appropriate selection of torsion angle conformations characteristic of the right- and left-handed α-helix (see Table 1 for whole classification of such conformations), the flat structure is retained. Figure 5B presents the mutual relations of the flat chain forms in a fibril structure. 

The discussion concerning the characteristics of Aβ (1-42) amyloids is included in Appendix A–Appendix A. 

Figure 5 presents the illustration of the proposed classification of conformations in the form of distribution of the Phi and Psi angles plotted on the maps of the radius of curvature R and the V angle.

The comparative analysis of the distribution of Phi and Psi angles on the Ramachandran map for the native and amyloid forms of transthyretin shows their concentration in the area of the maximum value of the radius of curvature and the V angle close to 180 degrees, as well as the increase in the number of points and their clear focus around the strictly defined α-helix conformation (right- and let-handed) (compare with Figure 5A,B). The exact characteristics of the distribution of Phi and Psi angles for the discussed structures revealing the concentration of angles around the line Psi =−Phi and in the areas of right (R)- and left (L)-handed α-helices are given in Table 2.

The summary (Table 2) presents the characteristics of the distribution of pairs of Phi and Psi angles in the mentioned specific conformations. A very small dispersion in the R- and L-α-helix conformations indicates the need for unambiguous orientation of the atoms building the hydrogen bond, providing a parallel system resulting in the presence of a hydrogen bond that does not disturb the planar system for the entire chain. The arrangement of the atoms in C=O… H-N in a straight line guarantees the greatest contribution to the stabilization. The energetical stability derived from the hydrogen bond depends on the angle between the C=O and H-N bonds. The maximum energetic benefit in the case of hydrogen bonds is to arrange the atoms that compose them in a rectilinear form. In the analysis of β structures, the E and F codes were distinguished. The distinction within the β structure of the E and F forms results from their different characteristics. The E code defines the β structures propagated in a straight line, and the code F refers to those components of the β form where there is a twist or a clear bend of the chain. The F conformations are those that terminate the propagation of the β form.

A significant concentration of the Phi and Psi angles in the case of β structure concerns primarily the conformation defined by the E code—meaning a rectilinear structuring of the polypeptide chain. The Phi and Psi angle dispersion for the β-structural area was determined as the average distance to the Psi =−Phi line in order to visualize the concentration of the angles in this area of the Ramachandran map. On the other hand, the values of the mean distances for the scattering measurement given in Table 2 express their locations in relation to the averaged position for the R- and L-α-helix.

In the detailed analysis, the local structures discussed in Table 1 show the orientation system of hydrogen bonds in parallel/antiparallel with the arrangement of atoms included in the hydrogen bond (Figure 6) maintaining the flat structure of the chain.

The 3D presentations of the fragments highlighted in Figure 4A, described in Table 1, and shown in Figure 6 reveal a characteristic system of hydrogen bonds. The black-white arrows visualize the pattern typical for the β-structural form (anti-parallel). Single residues with R- or L-α-helix conformation disturb the turn (if a vector is assigned to the hydrogen bond) while maintaining the parallelism of the hydrogen bonds (red arrows). Single residues with such a conformation do not disrupt the hydrogen bond system, maintaining the flatness of the single chain. The presence of a single amino acid with the R- or L- α-helical conformation results in a twist while maintaining the 2D chain structure. Similarly, the sequence of one R-α-helix conformation followed by one single residue of L-α-helix conformation introduces the form of twists keeping the 2D pattern of the form preserved. Successive hydrogen bond relationships do not disrupt the orientation despite the change in the direction of the hydrogen bonds.

An open question is the issue of the mechanism of obtaining such a set of conformations by amino acids representing completely different conformations in the form of a native protein.

### 2.2. The Relation of the Native Form to That Present in the Amyloid Fibril

The comparative analysis concerns two aspects: changes in the conformation of individual residues and the phenomenon of complexation as such.

#### 2.2.1. Change of Conformation

The characteristics of conformational changes resulting from a comparative analysis of the structure of native transthyretin and its amyloid form reveal the following rules. 

Items identified in their native form as AB with the torsion angles satisfying the relation Psi = −Phi are not the components of the segments classified as AB in amyloid.

The positions of single amino acids with conformation typical of the R-handed α-helix in amyloid are not those natively representing this conformation 

The conformations typical of the L-handed α-helix that appear are also the results of amino acid conformational changes. Neither amino acid of L-handed α-helix conformation is present in its native form.

Conformational changes tracked by the position of the Phi and Psi angles on the Ramachandran map are shown in Figure 7. The starting positions (native form) and the corresponding final (amyloid form) have been presented for those residues that obtain the conformation of L-handed α-helix in the amyloid (Figure 7A–C) as well as the change of the conformation of the R-handed α-helix in its native form to another conformation present in the amyloid form (Figure 7D–F). The objects of the analysis are the following structures of transthyretin: aggressively undergoing amyloid transformation (1G1O), resistant to this transformation (1GKO), and the native structure treated as the reference one (1DVQ).

All values of Phi, Psi angles were calculated taking the structures available in PDB (1DVQ—native form, 6SDZ—amyloid form). The amino acids were selected according to their L-α-helix conformation in amyloid and R-α-helix in native form of transthyretin.

The surprising conclusions resulting from the analysis of the distribution of the Phi and Psi angles in the native form concern the fact of changing the angles that meet the condition Psi = −Phi. This relationship is most often present in the form of amyloid (Figure 5). Amino acids with torsion angles satisfying such condition present in their native form in all structures discussed here change their conformation to that typical for L-handed α-helix (Figure 7A–C). On the other hand, some conformations typical for R-handed α-helix in their native form take a structure in the amyloid with Psi = −Phi (Figure 7D–F).

It should also be noted that the conformations typical of the L- and R-α-helices present in the structure do not imply the presence of chain sections representing this secondary structure. The clustering of points in the regions of the L- and R-handed α-helix results from the presence of individual scattered residues in the chain of amyloid form (Figure 8).

#### 2.2.2. Can the Amyloid Be Called IV-Order Structure?

In biology, proteins that reveal their biological activity only in the quaternary form are commonly present. Complexation can also be defined as the formation of a multi-chain fibril where the inter-chain interaction is present along the entire length of the polypeptide chain.

Assuming a single chain as a structural unit, there is a fundamental difference between the two constructs. Complexation leading to a quaternary structure preserves the three-dimensionality of the units constructing the complex system. Complexes and monomeric units show a spatial 3D structure. The components of the interfaces in the quaternary structures are selected amino acids (selected fragments of the polypeptide chain—often short or even individual from different parts of chain) with specific characteristics, which interact through side chains to stabilize the complex system.

The mechanism leading to the formation of the quaternary structure was discussed on the basis of the fuzzy oil drop model, which assumes that the protein folding process can be regarded as the formation of micelles. Such a formation aims at exposing polar residues on the surface and concentrating hydrophobic ones in the central part of the protein, showing that obtaining the 3D structure of a single chain relies essentially on obtaining a micelle structure with an exposed polar surface and centrally located hydrophobic residues [43]. Such a system guarantees the solubility in the water environment. The site and locus of complexation (interface zone) are often determined by a local deviation from the micelle-like system due to local exposure of hydrophobic residues. On this basis, the complex formed still represents a micelle-like system with a clearly three-dimensional structure [44].

The structure of the single chain in amyloid fibril is not micellized, since the individual chain represents a flat 2D structural form. The decisive factor in the construction of fibrils is the network of hydrogen bonds engaging almost all residues present in the chain. Each residue represents maximum involvement of C=O and H-N groups of peptide bonds of adjacent chains generating a strongly stabilized system. The structures of native transthyretin monomers and dimers show a micelle-like structure using the fuzzy oil drop model [45]. Such an observation is not present with regard to amyloid, including transthyretin amyloid in particular. 

Moreover, it is necessary to note the fundamental difference resulting from the present symmetry in the systems of the quaternary structure in relation to the structure of the amyloid fibril. The IV-order structure of complexes results from the symmetry operation, which is the axis of rotation (sometimes together with translation—see pilin). For transthyretin, this is the two-fold axis—C2 (Figure 9A). In contrast, the only element of symmetry transformation in amyloid is the translation operation (Figure 9B). The symmetry related to the axis of rotation provides an unambiguously anti-parallel system of β-structural sections that build a common β sheet. In the case of translation, the creation of a β sheet is possible only on the basis of a parallel β-strands system.

The treatment of the formed complexes (including dimers) with amyloid transforming proteins as precursors of this process is not warranted. The comparison of the results of the comparative analysis both regarding the presence of the appropriate conformation (change of α-helical forms into β-structural forms and vice versa) and the symmetry operation leading to the structuring of complexes suggest the conclusion that there is a necessary radical change in the entire system of the polypeptide chain. Reports about the need for a partial or significant degree of unfolding of the native protein turn out to be consistent with the analysis presented here.

The summary of conclusions resulting from this part of the analysis suggests that even the presence of the conformation expressed by the relation Psi = −Phi in its native form cannot be treated as the seed of the amyloid form. The appearance in the amyloid of a significant number of amino acids satisfying the Psi = −Phi relation is the result of a significant change in the structuring of the entire polypeptide chain. What additionally distinguishes the structuring of the fibril is the presence of only the translation operations as opposed to the axis of rotation, which determines the structuring in the oligomers. Obviously, the rotation axis type operation is present in amyloids with respect to the mutual orientation of proto-fibrils in the super-fibril structure, and thus in the case of amyloid forms available for Aβ (1-42) in 2MXU—protofibril [46], 2NAO—C2 two-fold axis [47], in 2MVX—C2 two-fold axis [48] and in 2MPZ—three-fold C3 axis [49]. The axis of rotation for these symmetry operations is the axis parallel to the axis of propagation of the superfibril.

In conclusion, “associate” seems to be the best term for amyloid fibrils. 

### 2.3. Modeling the Structural Changes According to the Elliptical Path from the Early-Stage Model of Protein Folding

The use of the elliptic path in the early-stage model as representing the limited conformational subspace to generate an early intermediate structure, and thus a highly unfolded structure, reveals the fundamental differences between the structuring of the reference transthyretin (1DVQ—Figure 10A), transthyretins: resistant to amyloid transformation (1GKO—Figure 10B) and aggressively undergoing this transformation (1G1O (Figure 10D) in relation to the amyloid form of transthyretin (6SDZ—Figure 10C). 

Figures shown in Figure 10 put together the early-stage forms generated using the ellipse path as limited conformational sub-space for the early step of folding with the native structures of the forms of transthyretin. The structures on the left side in each part of the figure represent the structures generated using Phie and Psie angles (Phie and Psie—mirrors of Phi and Psi angles after moving them toward ellipse path using the minimum distance criteria). On the right side of each picture, the native form is given. The amyloid structure is also presented in Phie and Psie representation to visualize possible pre-folding form. 

The marginally visible differences in the native 3D structures of the above-mentioned transthyretins, 1G1O, 1GKO and the reference 1DVQ, are revealed only after they are “transformed” to the state determined by the elliptical path. This can be obtained by projecting points (Phi and Psi torsion angle pairs) on the ellipse based on the principle of the shortest distance. The coordinates of these projections on the ellipse, denoted as Phie and Psie, define a chain conformation that can be regarded as reproducing the early stage of the folding process, which means a folding state with a predominant preference for the backbone itself (according to our two-stage folding model [32,33]). On the other hand, these structures can also be treated as the result of α-helix unfolding in the mentioned molecular dynamics experiment [38,39].

N-terminal fragment 11–35 (numbering according to 1DVQ PDB file—marked as blue marine in Figure 10) seems to represent in native forms a structure similar to that present in the amyloid—a hairpin. The difference lies in the loss of spatiality of this loop in native structures, which is flat in amyloid. After changing the conformation to the form expressed by the Phie and Psie angles, this flatness appears to a varying degree in the examples discussed.

The fragment 57–74 (marked as red in Figure 10) seems to undergo fundamental changes. It transforms from a short β-sheet form and a loose loop to a loop form in the case of 1DVQ and 1GKO, while in the aggressively transforming version, it becomes a stretched, almost β-strand segment similar to that present in amyloid. In the case of this section, the differences in the structures compared from the point of view of spatial construction are noticeable.

The fragment 74–102 (marked as green in Figure 10) preserves the α-helical segment that is absent in the amyloid form. In the case of this section, the main difference is the significant relaxation of the packed form present in native structures, where this section is an important component of the β-sheet.

A similar characteristic applies to the C-terminal segment (102–123—silver in Figure 10), which, being a β-sheet component in native form, transforms into loose segments that do not interact with any other part of the chain. If there is an interaction between the amino acid side chains of this segment in the amyloid form, then the β-sheet form is not involved in these detachments.

However, the main comparative assessment concerns the degree of loosening of the structure, which is most visible in the case of 1G1O. The structure determined on the basis of Phie and Psie angles in this case shows a significant relaxation consisting of the disappearance of any interactions within the chain, leading to the availability of the chain along its entire length for interactions that potentially allow the possibility of engaging the backbone in the new interactions.

In the Fuzzy Oil Drop (FOD) model (43), it is assumed that the structure containing the seed of the hydrophobic core is able to reproduce its full structure present in its native form. Assessing the early intermediary structures from the point of the presence of the hydrophobic core, the 1GKO protein retains the possibility of recreating the hydrophobic core in the form present in the WT version, while in the case of 1G1O, such a possibility does not exist [45]. This observation is assumed to cause the aggressive transformation leading to the formation of amyloid fibril by this protein. The unfolded structure according to the early-stage model carried out for the aggressive amyloid transformation does not show any form of hydrophobic core, which consequently may mean that the hydrophobic form present in the native form is not reproducible.

The two discussed proteins differ in the presence of five mutations. Three of them (tri-peptide) are located in the segment absent in the available structure of transthyretin amyloid (slightly visible white segment in Figure 10). The two remaining mutations, in the light of the analysis carried out here, seem to not play any significant structural role. Therefore, it is difficult to discuss the role of the tri-peptide mutation in the transformation process. Nevertheless, analysis of the remaining segments reveals significant differences, making the structure of the unfolded 1G1O most similar to the amyloid form, especially by making the backbone available along almost the entire length of the chain for potential new influences other than those present in the native form. 

A quantitative assessment of the degree of structural similarity of the examples discussed is currently under investigation and will be available in an independent study. Here, the discussion is focused on the foundations and rationale behind the introduction of the early-stage model as a model to reconstruct the intermediate structuring in the protein folding process.

### 2.4. The Proposed Model of Conformational Changes Leading to the Amyloid Form of Transthyretin

As already presented, it is assumed that the course of structural transformations follows the two proposed paths. Therefore, referring to the models described earlier, a procedure of conformational changes leading from the WT structure to the amyloid form can be stated in the following stages. 

Identification of the changes, i.e., the starting structure—target structure pairs—similarly as shown in Figure 8.

Determining the stepwise change in the Phi and Psi angles leading to the achievement of the Phie and Psie angles on the basis of the shortest distance to the Phi and Psi positions present in the WT form.

The Phie and Psie angles are changed stepwise along the elliptical path directing toward the final amyloid conformation.

After obtaining the conformation represented by the Phie and Psie angles satisfying the relation Psi = −Phi, the changes in the positions of the points representing the corresponding conformations are modified along the line Psi = −Phi.

When a target amino acid in the amyloid form reaches the conformation of torsion angles typical of L-handed α-helix, the changes proceed along an elliptical path leading to the region representing the L-handed α-helical forms.

For the conformational changes of right-handed α-helix, a path along the ellipse was proposed (L2 and G2 paths—Figure 2), as the conformational changes concern amino acids subjected to the constraints resulting from their connections to other parts of the chain—similarly to BPTI—that prevent the free movement of terminal residues.

If the target conformation approaches the line Psi = −Phi, the changes in the positions of the points (pairs of torsion angles) representing the corresponding conformations are directed along this line.

The presentation in Figure 11 aims to visualize the structural changes both from the point of view of the 3D assessment and the corresponding sets of Phi and Psi angles.

The model of conformational changes proposed above was applied to the fragment 11–35 of transthyretin and is shown in Figure 12 and Figure 13. The tracking of subsequent changes in the structure (shown in Figure 14) along with the corresponding changes in the positions of the points—the pairs of Phi and Psi angles—in accordance with the proposed model are shown in Figure 12. The changes in Phi and Psi angles followed the proposed procedure as: (1) approaching the ellipse, (2) move along the ellipse toward the β-structural area, and finally (3) for those amino acids that reach the line expressing Psi = −Phi, the changes satisfying this relation of the dihedral angles were performed. Each step of the procedure is shown in the form of a 3D structure (Figure 14), and appropriate changes in Phi and Psi angles are shown in Figure 13. The structural change analysis shown in Figure 12 reveals the significance of the structural change for step H → I, where a single different conformation (orange square) causes the observed difference in the 3D assessment (marked with red dash).

It should be noted that a large concentration of points around the line Psi = −Phi for the native form does not play the role of precursor for amyloid construction (Figure 7 and Figure 11). The presence of Phi and Psi satisfying the relation Psi = −Phi is the effect of structural transformations of other residues. 

The work on the entire structure of this protein requires the extension of the proposed model. The present work focuses only on the basic assumptions of a potential model recreating the amyloid transformation. Work focused on construction of the algorithm expressing conformational changes concerning the entire transthyretin molecule as well as other amyloid proteins (IgG light chain V domain—both the native structure and the amyloid forms) is currently underway. 

To verify the model presented with the application solely to transthyretin, the analysis of Aβ (1-42) amyloid forms, particularly the distribution of Phi and Psi angles, is presented in Appendix A–Appendix A. 

## 3. Discussion

The structure of amyloids has been an object of analysis for many years, providing new information on the determinants of the process of protein folding in its correct form. The protein folding process is closely related to the misfolding process. It is assumed that the search for a mechanism for the protein folding process will be supported by intensive works on the misfolding phenomenon due to the medical connotations of the latter process [50,51,52,53,54,55,56].

It is common to treat the transformation process as a “protein-only hypothesis”, meaning independence from chemical modification of the sequence present in the polypeptide chain, as evidenced by PrP studies, which can assume two isoforms in relation to the secondary and tertiary structures [57,58].

Based on the analysis performed in this work, the following observations can be made.

There is a need for a significant degree of unfolding of the native structure prior to obtaining the amyloid structure through experimental determination of partially denaturing conditions (particularly in lysozyme) with lower pH [59,60,61,62,63,64]. Some of the works indicate the importance of the role of the backbone behavior [65,66]. 

Significant dependence on the environment—apart from the conditions such as pH, temperature, and ionic strength, there are also conditions related to the presence of the so-called chaotropic agents [67]. The phenomenon of the dependence of the folding process on environmental conditions was presented in the works [45,68] in which the possibility of restoring the hydrophobic core in the transthyretin resistant to amyloid transformation and aggressively undergoing amyloid transformation were investigated. The next example is the analysis of protein’s membrane [44,45,67] structuring, where a radical change in external conditions (the hydrophobic environment of the membrane) determines the system adjusted to the conditions different from that of the polar water. The role of the environment has been indicated as a factor that directs both the misfolding and folding processes [56,69,70]. Significant importance is also attached to the recently recognized unfolding protein response (UPR) mechanism in maintaining protein folding homeostasis [56,71,72].

Different from other reports is the assumption concerning the precursors of the formation of amyloid fibrils—polar zippers [73,74,75,76,77,78,79]. For example, in [31,80], β-strands present in amyloid alpha-synuclein, Aβ or tau are treated as precursors for amyloid transformation; in [81], misfolding and aggregation in proteins is connected with intrinsically disordered regions. 

The significant role of hydrogen bonding networks resulting in a pattern of amyloid fibrils in the form of β-sheet conformation, in which the hydrogen bonding direction runs parallel to the fiber axis and the β-strands are perpendicular, much like the rungs of a ladder, has already been discussed in [82]. 

The methods of complexation mentioned as structures in the anti-parallel β-sheet system seem to be valid only for super-fibril structures [83]. 

The characteristic features of amyloid structuring in the form of the presence of conformations typical of L-handed α-helices analyzed in the present study were also indicated in [83].

In the present work, the amyloid structuring was analyzed on the basis of the two-stage folding model using the early intermediate model, i.e., by significantly unfolding the chain present in the WT form.

## 4. Materials and Methods

### 4.1. The Early Stage Model of the Protein Folding Process

The early stage model of protein folding is based on the assumption that the backbone solely decides the structure of a polypeptide chain. It is based on the simplified representation of a polypeptide chain using only two geometrical parameters: the radius R of curvature of chain fragments and the angle V (the angle between the two consecutive peptide bond planes).

Here, we only sketch a general scheme for constructing this model (the detailed description is in [32,33]). 

The structures of alanine pentapeptide were generated for each point on the Ramachandran map (with 5 degree steps) and properly oriented in the common coordinate system (the averaged positions of C and O atoms of C=O groups determined the Z axis). The projection of Cα atoms on the XY plane allows the calculation of radius of curvature R for each pentapeptide as well as the V angle (for details, see [32,33]). The radius of curvature R appeared to be dependent on the orientation (the angle V) of consecutive peptide bond planes. For example, the V angle is equal to 0 deg for parallel orientations of peptide bond planes in α-helix. The V angle reaching 180 deg is present in β-structural form where the radius R is theoretically infinitely large due to linear construction of β structure (Figure 14A,B). The value of the V angle is the immediate consequence of Phi and Psi rotations. All intermediate structural forms represent the gradual increase in radius (starting at α-helical form) as dependent on gradually increased V angle. The dependence of R (logarithm scale) on the V angle is shown in Figure 14C with the parabolic curve superimposed. This curve was determined by the approximation of points representing low-energy areas on the Ramachandran map shown in Figure 14D (it is assumed that the relaxed form of the chain for a given V angle is represented by an appropriate radius of curvature). Figure 14E presents the elliptical path determined on the basis of points expressing the dependence of ln R on the angle V (for details, see [32,33]).

**Figure 14 ijms-23-09502-f014:**
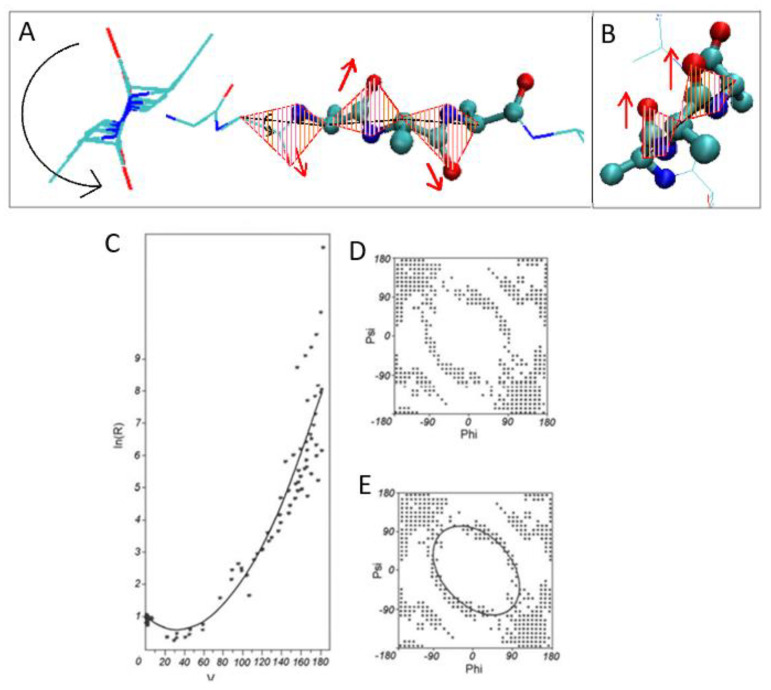
The main steps for generating the early-stage model (see the description in text). (**A**)—β-structural form with peptide bond planes marked by red lines visualizing the antiparallel orientation of C=O groups in two sequential residues. Left picture with the arrow visualizes the V angle = 180 degrees. (**B**)—α-helix with peptide bond planes marked by red lines visualizing the V angle = 0 degrees. (**C**)—parabolic dependence of R (logarithm scale) on V angle. (**D**)—distribution of points satisfying the condition defined by the parabola in (**C**). (**E**)—the elliptical path.

According to information theory, the chain with conformation “on the ellipse path” is balanced with the amount of information carried solely by the amino acids in the chain [84,85]. Such an elliptical structure can play the role of good starting point for the next step of the folding process when interactions are included and was described in our late-stage FOD model—the environment is the source of the lacking portion of information between the information carried by amino acids and required to define particular Psi angles in complete conformational space.

The elliptical path allows the introduction of the so-called structural codes. We denote the shortest path’s projection of points on the Ramachandran map (i.e., the pairs of Phi and Psi angles) from native protein onto the elliptical path by a pair of Phie and Psie angles (index “e” denotes belonging to the elliptical path). The frequency profile of Phie and Psie angles from all proteins in the non-redundant PDB database reveals the presence of seven local peaks as the basis for our definition of structural codes A–G (for details, see [85]). 

The importance of the E and F zones in amyloid transformation is discussed in Appendix A–Appendix A. 

### 4.2. Programs Used 

Potential users have the following two possibilities to access the program used in our calculations. 

The first option is a program allowing calculation of hydrophobicity profiles and the parameter RD accessible upon request on the CodeOcean platform: https://codeocean.com/capsule/3084411/tree, accessed on 15 July 2022. Please contact the corresponding author to gain access to your private program instance. 

The second option is the application implemented in collaboration with the Sano Centre for Computational Medicine (https://sano.science, accessed on 15 July 2022) and running on resources contributed by ACC Cyfronet AGH (https://www.cyfronet.pl, accessed on 15 July 2022) in the framework of the PL-Grid Infrastructure (https://plgrid.pl, accessed on 15 July 2022) that provides a web interface for the abovementioned elements and is freely available at https://hphob.sano.science, accessed on 15 July 2022.

The VMD program was used to present the 3D structures. 

[https://www.ks.uiuc.edu/Research/vmd/—accessed on 16 May 2022].

## 5. Conclusions

The peculiarities of structuring the polypeptide chains present in amyloids (here, the discussion was limited to transthyretin and Aβ (1-42) in Appendix A–Appendix A) reveal requirements that favor the formation of an amyloid fibril structure. Providing a flat structure for an individual chain is a result of the presence of a specific set of Phi and Psi angles ensuring the presence of anti-parallel orientations of hydrogen bonds derived from the groups C=O and H-N of peptide bonds. The β structure with angles meeting the condition Psi = −Phi (for Phi < 0 and Psi > 0) provides a perfectly anti-parallel system of hydrogen bonds. The presence of single amino acids with an R-α-helical or L-α-helical conformation maintains this type of order of the hydrogen bonding system by introducing only a parallel system for the hydrogen bond turn. Such a system enables the involvement of (almost) all amino acids (peptide bonds) in stabilization based on hydrogen bonds, which are the main factor of energy stabilization. However, this system also imposes a parallel arrangement for adjacent chains. This is because only such a system ensures the complementarity in the system of turns in hydrogen bonds.

The specificity of the structure of the amyloid fibril also lies in the ordering based on the operation of symmetry, which is solely the translation. This factor significantly differentiates the structuring of the fibril from the quaternary structuring, where the symmetry operation is an axis of rotation (appropriate multiplicity of axes) or the combination of the axis of rotation with translation (for example, the structure of pilin). The limitation to the translation operation results in the presence of only the parallel orientation for the β-sheet construction versus the anti-parallel orientation as a result of the monomer unit rotation operation.

The structural transformation leading to the formation of the fibrous form requires, in the light of the analyses presented here, a radical change in the conformation of individual residues as well as in the mutual orientation of individual amyloid fibril chains. The indicated need for partial unfolding of proteins undergoing amyloid transformation [59,60,61,62,63,64], or also significant protein unfolding [59,60,61,62,63,64], has been proposed. In the present work, we based our results on the simulations of molecular dynamics performed by Daggett and Levitt [38,39] leading to the unfolding of α-helical sections. These results appeared consistent with the conclusions based on the early-stage model introduced by the authors of this paper [32,33]. These two models linked together may be applied to the simulation of misfolded proteins. It is assumed that the mechanism for these two processes is common.

Studying the amyloid transformation of transthyretin for the whole molecule requires extensive software development, which is under preparation. Here, we limited the description of the basics leading to the further, broader model of amyloid transformation.

Currently, only the significant structural differences have been demonstrated in the form of partially unfolded transthyretin in its two forms: aggressive and resistant to amyloid transformation. The degree of similarity of the structure of the aggressive form with that present in the amyloid fibril seems to provide arguments for the introduced model.

For the analysis of the spatial structure of proteins, a fuzzy oil drop model was introduced, with the external field from the aquatic environment and its modified FOD-M form taking into account the influence of extra-water environmental factors on the structuring of proteins, including membrane proteins in particular. The participation of factors expressed in the model by the value of the parameter K seems to be a universal model of the same type as that for chaotropic agents [67]. The type of factor that would favor the formation of an appropriate system of hydrogen bonds, the presence of which seems to be the dominant factor in the stabilization of amyloid structures, is being sought.

For the amyloid transformation, a form of external force field (water structuring) is sought that prefers and strongly supports structural constructions based on hydrogen bonds. It seems that the key to the search for a solution to the mechanism of any structural transformation of proteins lies in the appropriate structuring of the aquatic environment itself. The structure of water remains unrecognized [86,87,88,89,90,91,92,93,94,95,96,97,98,99,100,101], and it plays a critical role for the entire world of biological processes, directing them in a way that ensures the proper functioning of all biological systems. Searching for a mathematical model of a specific external force field, which is water, is important especially in the context of a method favoring the formation of amyloids in the form of shaking, which increases the presence of the air–water interface [100]. The possible extension of the FOD model, including the representation of external factors supporting the generation of the hydrogen bond network, is currently under consideration. 

It should be underlined that the presented model is of highly hypothetical character. However, even this introductory form of the model may be applied in simulations of the protein folding process, including also the misfolding process. 

## Figures and Tables

**Figure 1 ijms-23-09502-f001:**
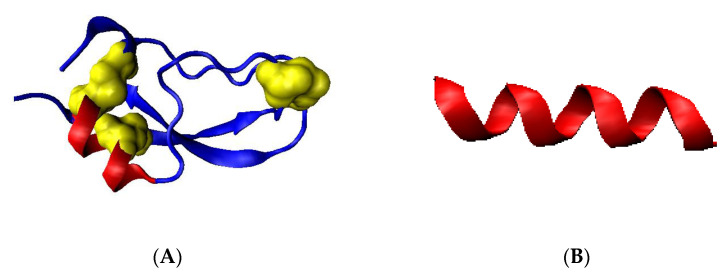
Presentation of 3D structures. (**A**)—the BPTI protein with highlighted helical section (red) and disulfide bonds (yellow)—PDB ID 4PTI. (**B**)—the chain of polyalanine (13 amino acids) in α-helical form.

**Figure 2 ijms-23-09502-f002:**
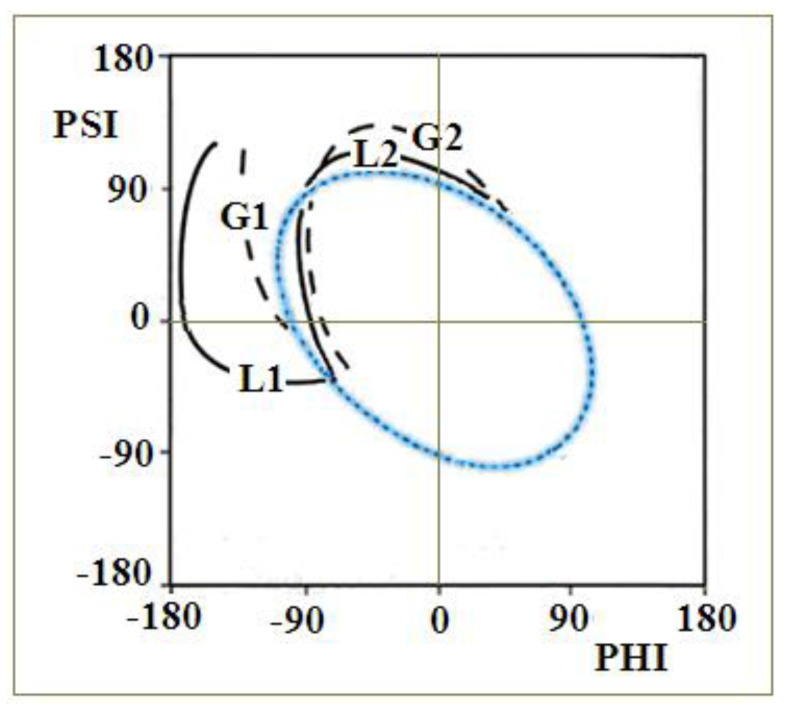
Two paths determined on the basis of the distribution of Phi and Psi angles obtained by molecular dynamics [38,39]. Line L1—unfolding of the α-helix with increasing radius of curvature of the chain obtained for unfolding of the free α-helix, Line L2 and G2—structural changes leading to unfolding of the α-helix as a component of the BPTI structure by reducing the radius of curvature. Line G1—an alternative route for the unfolding of the α-helix as a component of a longer polypeptide chain in a protein. The blue line (ellipse) is the optimal path for structural changes as identified using the early-stage model described in Materials and Methods.

**Figure 3 ijms-23-09502-f003:**
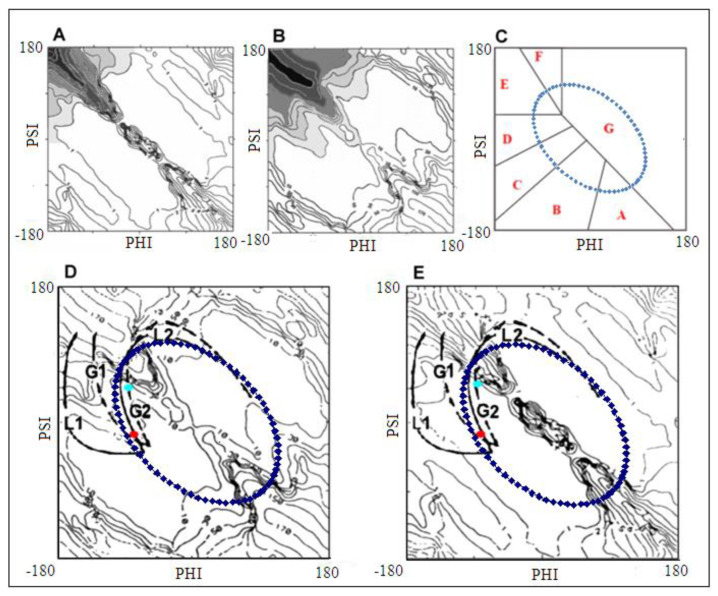
(**A**,**B**)—the distribution of geometrical parameters R, V in early-stage model, (**C**)—the zones corresponding to the structural codes and the elliptical path, (**D**,**E**)—the L, G and ellipse paths superimposed on V and R distributions, respectively. Red points—helix 3_10_, turquoise point–helix 2_7_.

**Figure 4 ijms-23-09502-f004:**
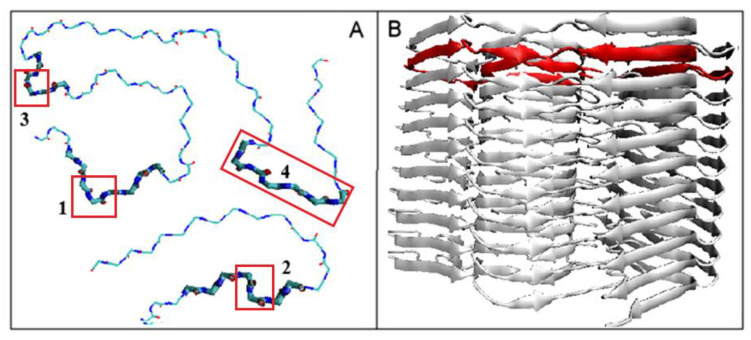
The structure of transthyretin chain in amyloid fibril. (**A**)—single chain (6SDZ)—sections maintaining a flat chain arrangement with specific non-β-structural conformations are highlighted (see their classification in Table 1). (**B**)—amyloid fibril with one chain marked in red for better visualization. The distinguished red chain in (**B**) is shown in (**A**) section in another perspective. The aim of both presentations is to show the planarity of the chain structure.

**Figure 5 ijms-23-09502-f005:**
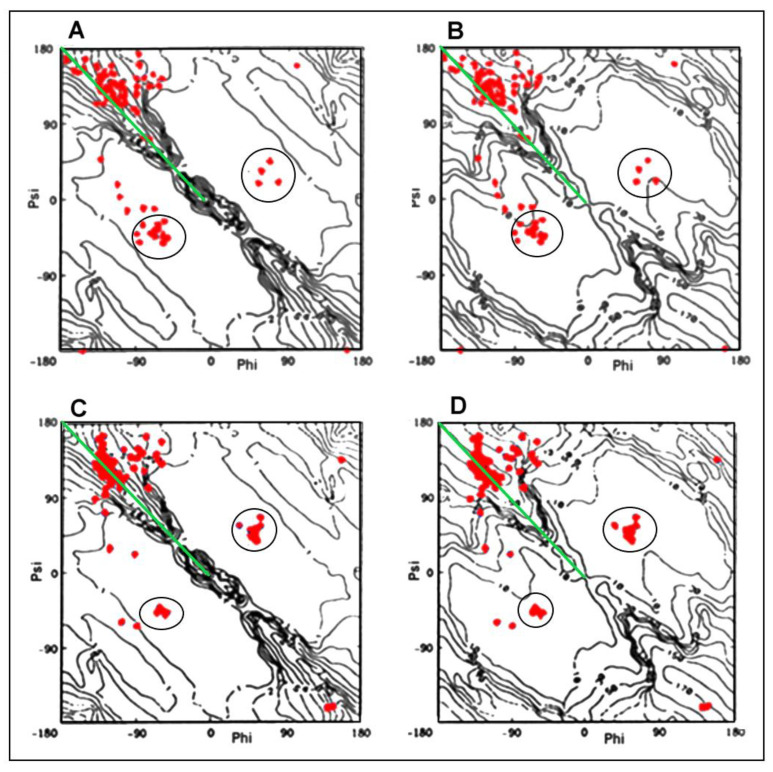
Superposition of pairs of Phi, Psi angles for transthyretin—reference form 1DVQ (**A**,**B**) and amyloid form—6SDZ (**C**,**D**) (red points): (**A**,**C**)—on radius of curvature map; (**B**,**D**)—V-angle map. Green line visualizes the position of Psi = −Phi. The approach to this line can be seen in amyloid forms in respect to the distribution observed in WT form of transthyretin. The blue circles visualize the low distribution of angles in right- and left-handed α-helix in amyloid form in respect to analogous WT forms of transthyretin.

**Figure 6 ijms-23-09502-f006:**
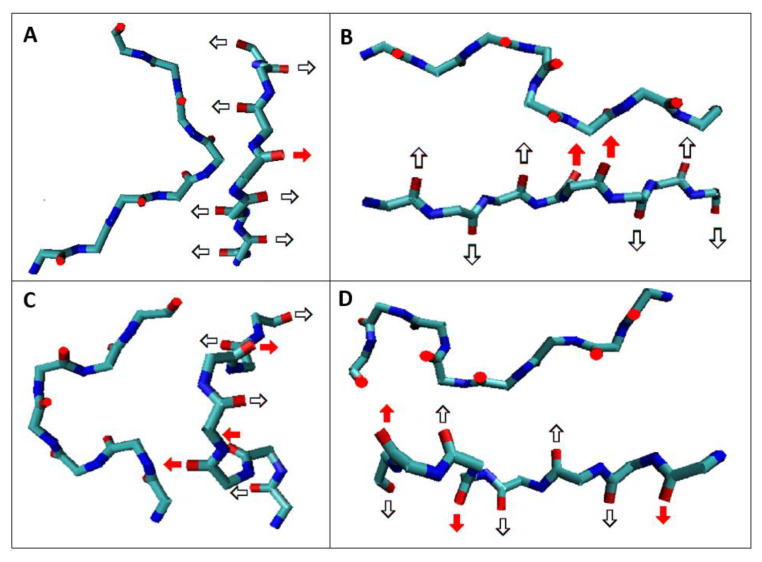
Examples of orientation of hydrogen bonds in fragments of flat structure of polypeptide chain in amyloid as listed in Table 1 in the absence of β-structural forms: (**A**)—(fragment 1 in Figure 4A) AB-L-AB; (**B**)—(fragment 2 in Figure 4A) AB-17L-18R-AB; (**C**)—(fragment 3 in Figure 4A) AB-87L-88R-88B-89L-90B-AB; (**D**)—(fragment 4 in Figure 4A). AB-105M-106L-107B-108AB-109L-110B-111L-112B-113AB. Black and white arrows—orientation resulting from β structures introducing antiparallel system; red arrows—current R or L-α-helix conformation leading to a change of turn while maintaining parallel orientation. Two different spatial orientations of the discussed sections are shown.

**Figure 7 ijms-23-09502-f007:**
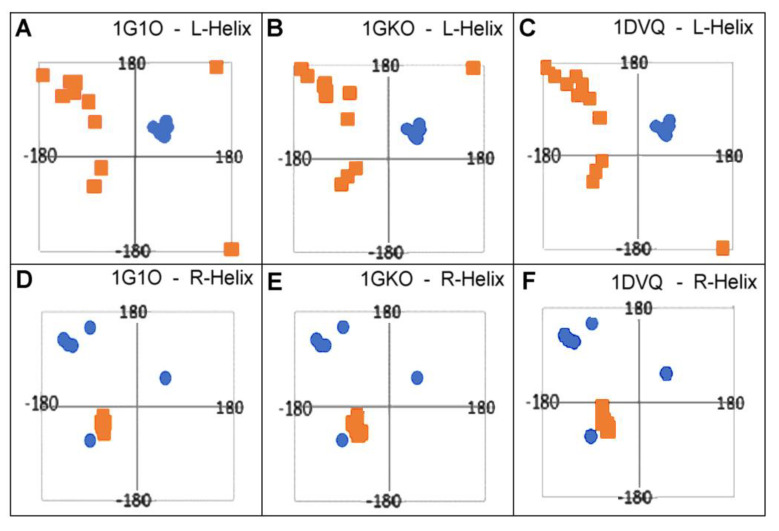
The migration of Phi, Psi angles as compared in native and amyloid forms of transthyretin. (**A**–**C**)—The amino acids of L-handed α-helix conformation in amyloid (blue circles) migrated from the positions shown as orange squares in native form. (**D**–**F**)—The amino acids of R-α-helix in the native form (orange squares) migrated to positions in amyloid (blue circles).

**Figure 8 ijms-23-09502-f008:**
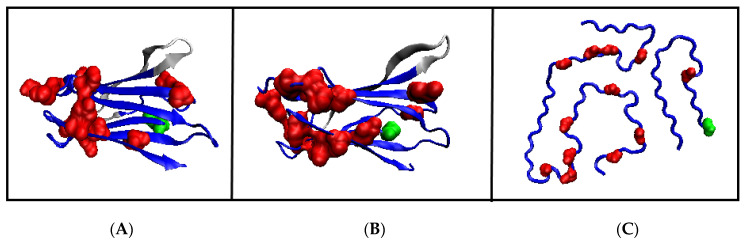
3D structures of transthyretin. (**A**)—transthyretin resistant to amyloid transformation—1GKO, (**B**)—aggressively undergoing amyloid transformation 1G1O, and (**C**)—amyloid form 6SDZ. Red residues and those representing the L-α-helix conformation in amyloid form are marked. White fragment—a section of the chain absent in the amyloid structure. Green residue—N-terminal position (for easy navigation).

**Figure 9 ijms-23-09502-f009:**
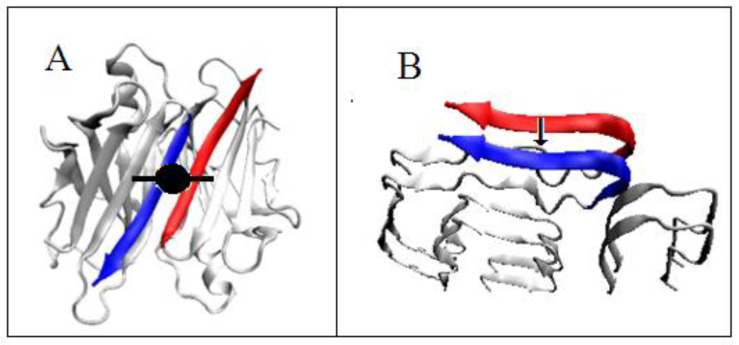
The structure of transthyretin. (**A**)—native homo-dimer—the double axis expressing operation of symmetry marked. (**B**)—two chains in the amyloid form—the arrow shows the translation operations. The sections responsible for dimer stabilization by creating a β sheet propagating through both monomeric units in an anti-parallel form are marked. The same fragments in the amyloid system represent the β-structure system in parallel form.

**Figure 10 ijms-23-09502-f010:**
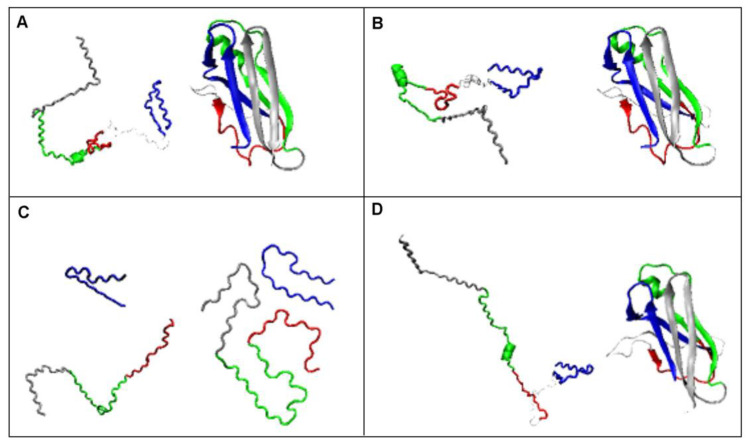
Presentation of transthyretin structures. (**A**)—1DVQ—reference, (**B**)—1GKO—resistant, (**C**)—6SDZ—amyloid, (**D**)—1G1O—aggressive. Analogous fragments of the chain in the appropriate colors: N-terminal section—blue marine, C-terminal section—silver. Thin, marginally visible white section—fragment 36–56 not present in the amyloid structure.

**Figure 11 ijms-23-09502-f011:**
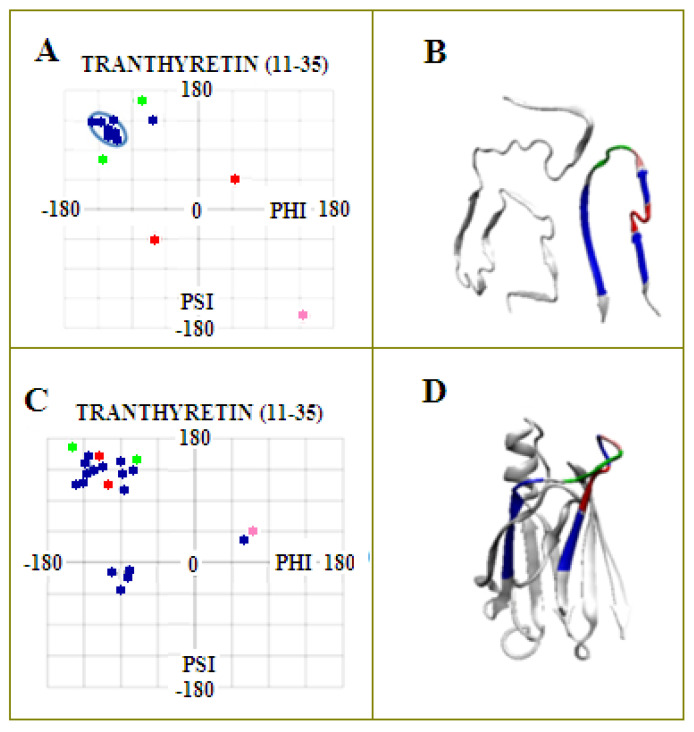
Fragment 11–35 in the structure of the amyloid (shown in **A**,**B**). (**A**)—distribution of the Phi and Psi angles present in this fragment. (**B**)—3D structure with a highlighted (navy blue) section being the object of the analysis, with the sections highlighted in red—the place of disturbance of the rectilinear β-structure. The set of angles fulfilling the condition Psi = −Phi in the native form (shown in **C**,**D**) was distinguished. Two red points (**A**) correspond to the conformation typical for left and right-handed α-helix causing a zigzag—red in Figure B; two green points—proline and adjacent serine; pink point—glycine forming a twist in the upper part of the loop. (**D**)—3D presentation of the structure of the transthyretin chain in amyloid form with highlighted fragments according to the color scheme in (**C**). A set of navy blue points—a model set of Phi angles in the mutual relation Phi = −Psi creating a rectilinear structure—navy blue fragments in (**B**). Colors consistent with the 3D presentation and the respective Phi and Psi angles.

**Figure 12 ijms-23-09502-f012:**
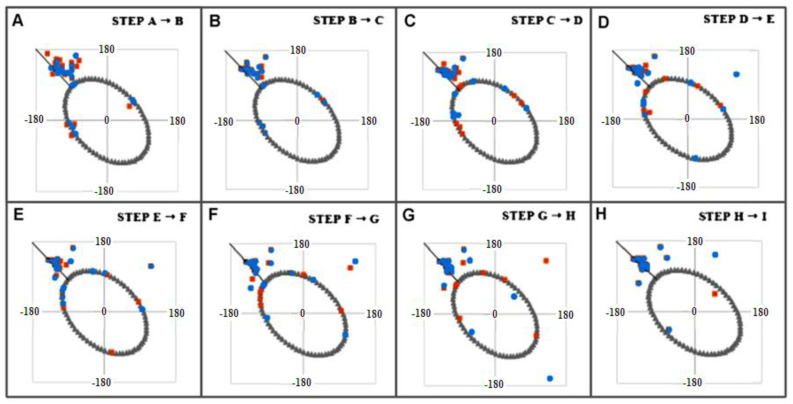
Step-wise changes in Phi and Psi angles following the procedure described above in the text. The red dots represent the conformation of the starting structure—the post-transformation one. The 3D presentation of the appropriate structures shown in Figure 13. Notice that one single Psi difference in the structure H is shown in Figure 13 as red dash on structure H. The gray ellipse—the elliptical path treated as conformational sub-space for early stage of folding process. The captions in this Figure follow the captions given in Figure 13.

**Figure 13 ijms-23-09502-f013:**
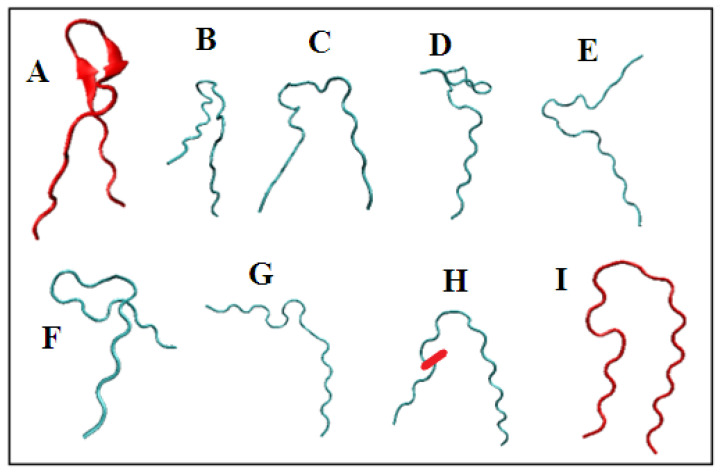
The proposed model applied to the fragment 11–35 of transthyretin (1DVQ). (**A**,**I**)—the structures as observed in 1DVQ and 6SDZ, respectively, shown in red. (**B**–**H**)—step-wise changes in structure according to scenario described above in the text. The stepwise changes in the Phi and Psi angles are shown in Figure 12. The red dash on H—the single position of Phi and Psi angles different in respect to the final structure (**I**). The difference shown in Figure 12H.

**Table 1 ijms-23-09502-t001:** The proposed classification of specific structural forms leading to maintenance of a planar structure of chains present in amyloid fibrils. β-structure conformations are marked in blue-marine, with those different from β-structure in red. First column (left) lists the identification as shown in Figure 4A. Symbol designations: AB—amyloid form of β structure (Phi and Psi angles fulfill the relation Psi = −Phi), R/L—Phi and Psi angles from right/left-handed α-helix area of Ramachandran plot, B—β structure, M—Phi, Psi angles from the area between α-helical and β structural forms. The numbers accompanying the codes—positions of residues in the fragments under consideration.

NR	Codes of Specific Conformation	3D Structure of Conformation
1	AB-63L-AB	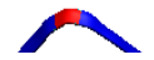
2	AB-17L-18R-AB	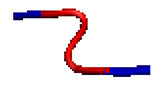
3	AB-87L-88R-88B-89L-90B-AB	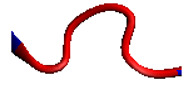
4	AB-105M-106L-107B-108AB-109L-110B-111L-112B-113AB	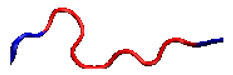

**Table 2 ijms-23-09502-t002:** Summary of the mean distances for the above-mentioned forms of the secondary structure in the analyzed proteins. The average distance for the R- (right-handed) and L- (left-handed) α-helix is calculated in relation to the averaged position of the Phi and Psi angles in a given structure. The average distance for the β-structural area is relative to the line Psi = −Phi.

	R-α-Helix	L-α-Helix	β-Structural Area		
			E + F	E	F
6SDZ	4.84	7.80	14.77 (64)	9.65 (56)	50.60 (8)
1DVQ	24.05	12.10	19.95 (70)	13.18 (58)	52.67 (12)
1GKO	35.18	14.91	19.37 (74)	13.43 (62)	50.08 (12)
1G1O	14.80	13.27	20.98 (78)	13.64 (64)	54.56 (14)

## Data Availability

All data can be available on request addressed to the corresponding author. The program allowing calculation of RD is accessible on GitHub platform: https://github.com/KatarzynaStapor/FODmodel, accessed on 15 July 2022, and on platform https://hphob.sano.science, accessed on 15 July 2022.

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
