# Peer review of "The Possible Mechanism of Amyloid Transformation Based on the Geometrical Parameters of Early-Stage Intermediate in Silico Model for Protein Folding"

_ijms, 2022, doi:10.3390/ijms23169502_

Round 1
Reviewer 1 Report
Amyloid formation is a very hot topic and affects aspects of protein folding and public health. The current manuscript uses transthyretin amyloidosis (ATTR), a heritable amyloid disease that can also occur spontaneously, as a study subject. The (mis)folding is examined by molecular dynamics simulations, which are unfortunately poorly described in detail There are a number of interesting new findings in this manuscript that are of interest to a broader community of researchers working in the fields of protein folding and amyloidosis-related diseases.
The manuscript suffers from flaws in presentation and should be extensively rewritten, especially also in the Methods section.
The figures could also need improvement, e.g. Fig. 5b that is low-resolution and difficult to read. The V-angle should be shown in a figure as well: I think I have understood what is meant but this would help readers a lot in understanding.
Figure legends also need improvement, for example Fig. 11 - the caption is meaningless and to me it is not clear what is legend and what is main text??
Phis/Psi angles are often spelled 'Phie' or 'Psie', e.g. pages 17 to 20.
The meaning of'left-handed helix' should specified - it are single residues in this conformation, there are no longer stretches in the amyloid structure 6SDZ that have this conformation. For the sake of clarity, 'helix' should be written as ?-helix since a β-structure is helical as well! And the 'Beta-structure' should be consistently written as 'β-structure'! (I do not know why it is capitalised anyway).
There are a number of lingual glitches , e.g. on page 7 ... ' is able to maintenance of parallel system of hydrogen bonds.'
Formatting of the manuscript should be improved as well - there are a number of very abrupt changes in font size and in the reference section sometimes the authors are underlined and sometimes the titles - I am sure there are instructions and word processing templates available.
A carefully rewritten manuscript most likely would deserve publication in IJMS.
Beside the formal shortcomings, a major problem are the ill-described MD conditions.
Author Response
REVIEWER I
Open Review
(x) I would not like to sign my review report
( ) I would like to sign my review report
English language and style
( ) Extensive editing of English language and style required
(x) Moderate English changes required
( ) English language and style are fine/minor spell check required
( ) I don't feel qualified to judge about the English language and style
|
Yes |
Can be improved |
Must be improved |
Not applicable |
|
|
Does the introduction provide sufficient background and include all relevant references? |
(x) |
( ) |
( ) |
( ) |
|
Are all the cited references relevant to the research? |
(x) |
( ) |
( ) |
( ) |
|
Is the research design appropriate? |
(x) |
( ) |
( ) |
( ) |
|
Are the methods adequately described? |
( ) |
( ) |
(x) |
( ) |
|
Are the results clearly presented? |
( ) |
(x) |
( ) |
( ) |
|
Are the conclusions supported by the results? |
( ) |
(x) |
( ) |
( ) |
Comments and Suggestions for Authors
Amyloid formation is a very hot topic and affects aspects of protein folding and public health. The current manuscript uses transthyretin amyloidosis (ATTR), a heritable amyloid disease that can also occur spontaneously, as a study subject. The (mis)folding is examined by molecular dynamics simulations, which are unfortunately poorly described in detail There are a number of interesting new findings in this manuscript that are of interest to a broader community of researchers working in the fields of protein folding and amyloidosis-related diseases.
The manuscript suffers from flaws in presentation and should be extensively rewritten, especially also in the Methods section.
The Methods section describing the main idea of early stage model has been completely rewritten . Figure 3 has been extended – the mutual orientation of peptide bond planes for β-structure and α-helical form has been added.
The figures could also need improvement, e.g. Fig. 5b that is low-resolution and difficult to read. The V-angle should be shown in a figure as well: I think I have understood what is meant but this would help readers a lot in understanding.
The legend of Figure 5 – changed – currently Figure 4.
Figure legends also need improvement, for example Fig. 11 - the caption is meaningless and to me it is not clear what is legend and what is main text??
The detailed description of Fig 11 (currently Fig. 10) added to the text.
Phis/Psi angles are often spelled 'Phie' or 'Psie', e.g. pages 17 to 20.
The Phie and Psie are used intentionally. We denote the shortest path’s projection of points of Ramachandran map (i.e. the pairs of Phi, Psi angles) from native protein onto the elliptical path by a pair of Phie and Psie angles (index “e” denotes belonging to the elliptical path). The definition has been included in section 4.1 (Methods).
The meaning of'left-handed helix' should specified - it are single residues in this conformation, there are no longer stretches in the amyloid structure 6SDZ that have this conformation.
The presence of isolated residues of R- or L-α-helical are present in amyloid form of transthyretin (6SDZ). Together with β-structural forms produce the flat structure of the entire chain (Figure 5).
For the sake of clarity, 'helix' should be written as ?-helix since a β-structure is helical as well! And the 'Beta-structure' should be consistently written as 'β-structure'! (I do not know why it is capitalised anyway).
CORRECTED
There are a number of lingual glitches , e.g. on page 7 ... ' is able to maintenance of parallel system of hydrogen bonds.'
CORRECTED
Formatting of the manuscript should be improved as well - there are a number of very abrupt changes in font size and in the reference section sometimes the authors are underlined and sometimes the titles - I am sure there are instructions and word processing templates available.
Sorry for that – we did our best to correct it.
A carefully rewritten manuscript most likely would deserve publication in IJMS.
The text got significantly changed and extended. We focused the attention on making the text more clear for Readers. All modified fragments are given in red in the revised form of paper.
Beside the formal shortcomings, a major problem are the ill-described MD conditions.
The MD simulation is mentioned in the paper in form of citation of the results received by Daggett and Levitt [38,39]. The simulation of α-helix unfolding in increasing temperature revealed the path of conformational changes starting from α-helix and leading to β-structural and even in high temperaturÄ™ to L-α-helix. The path revealed by this simulation appeared to be consistent with the path defined by the two-steps model for protein folding developped by Authors of the current paper. This is why no information describing the MD simulation is given in the current paper.

Reviewer 2 Report
The work presented is particularly interesting because of its subject. It is also quite complex to follow, because it is based on a little old literature on the one hand and on a rather complex approach on the other. I was particularly interested in reading it, but because of its length and I sometimes had trouble following the authors' thoughts
The introduction is quite long, because it only presents the questioning of amyloid on one side, the published dynamics approaches and the model used. On this part, questions arise about the rather old work on the BPTI protein. Is there no more recent work, new structures, more complete approach or even the possibility of redoing it with recent methodologies?
Unfortunately, I did not understand figure three, how is this research produced? Is it really FOD, or is it dynamic? It would be good to specify this better.
The definition of the V angle should be detailed.
Figure four. Turkus' must be a Polish term.
How is table 1 proposed, and on what criteria?
It is not clear where figure 5A comes from.
Table 2 I don't understand the E+F.
How is the twist or bend calculated?
How is the set of examples shown in figure 7 obtained? The details are not precise enough to be sure of the methodology?
Figure 8 the red should be the orange. Do all the points move between R and L, or is it just an impression?
When you talk about the condition Psi = - Phi
Does this mean that 70 becomes -70, or just that the positive values become negative?
When you see the figure 13, it is normal to wonder if the resistant form and the aggressive form are not the same. Is there really a structural difference?
I didn't understand the discussion about complexes and dimers. Are we sure of anything?
Section 2.5 is surely the most relevant and expected. The FOD approach seems to be particularly appropriate for understanding this type of biological property.
Section 2.6 is a bit more difficult and especially the changes in Phi and PSi values are not clear. Is this just a follow-up of the proposed L2 and G2 lines?
In figure 14 the question arises as to the number of residues on the ellipse. Is it significant or very punctual?
The part on Abeta and well stretched obligatory in view of the subject but because of the length could go into additional material.
The discussion is a bit disappointing. One would like to have a more critical view of the approaches used and the results proposed. Also, more thought should be given to the importance of the sequence in the process; is it taken into account in a fine enough way so that mutants with drastic Clear impacts can really be taken into account.
The material and method section is too short. One would like to have more intermediate details on the different calculations made.
So this paper is very interesting, but particularly long and it is sometimes difficult to know what messages each section brings. It would be good to reduce it a bit and to highlight more relevant results to help the reader to read it properly.
Author Response
REVIEWER II
Open Review
(x) I would not like to sign my review report
( ) I would like to sign my review report
English language and style
( ) Extensive editing of English language and style required
( ) Moderate English changes required
( ) English language and style are fine/minor spell check required
(x) I don't feel qualified to judge about the English language and style
|
Yes |
Can be improved |
Must be improved |
Not applicable |
|
|
Does the introduction provide sufficient background and include all relevant references? |
( ) |
(x) |
( ) |
( ) |
|
Are all the cited references relevant to the research? |
(x) |
( ) |
( ) |
( ) |
|
Is the research design appropriate? |
(x) |
( ) |
( ) |
( ) |
|
Are the methods adequately described? |
( ) |
(x) |
( ) |
( ) |
|
Are the results clearly presented? |
( ) |
(x) |
( ) |
( ) |
|
Are the conclusions supported by the results? |
( ) |
(x) |
( ) |
( ) |
Comments and Suggestions for Authors
The work presented is particularly interesting because of its subject. It is also quite complex to follow, because it is based on a little old literature on the one hand and on a rather complex approach on the other. I was particularly interested in reading it, but because of its length and I sometimes had trouble following the authors' thoughts
The introduction is quite long, because it only presents the questioning of amyloid on one side, the published dynamics approaches and the model used. On this part, questions arise about the rather old work on the BPTI protein. Is there no more recent work, new structures, more complete approach or even the possibility of redoing it with recent methodologies?
Unfortunately, I did not understand figure three, how is this research produced? Is it really FOD, or is it dynamic? It would be good to specify this better.
Figure 3 is connected with our early stage model. Its short description in Methods section has been corrected to enable better understanding. All newly added fragments of the text are given in red.
The definition of the V angle should be detailed.
The general definition of V-angle as the angle between the two consecutive peptide bond planes has been added. Instead of the long, detailed definition of V-angle and R-radius of curvature (described already in previous separate papers), they were introduced in a graphical form to capture their idea. We hope this form is clear and opens the imagination.
Figure four. Turkus' must be a Polish term. CORRECTED
How is table 1 proposed, and on what criteria? –
The criteria are shown on Fig. 5A. The selection – as it may be seen – distinguishes the forms which are non-linear. Different forms of bends are taken for detailed analysis.
It is not clear where figure 5A comes from. –
Fig. 5A and B. – structures taken from PDB (6SDZ) Fig.5B. – the complete fibril with chain B distinguished in red. Fig.5A. – the chain B from Fig.5.B in orientation co-planar with the plane of paper. Currently the number of this Figure is 4.
Table 2 I don't understand the E+F.
How is the twist or bend calculated?
Zones E and F are explained on the Ramachandran map added to Fig. The detailed explanation is given in Supplementary Materials.
Explanation is as follows:
The Phi, Psi angles of all proteins present in PDB (non-redundant base) moved toward ellipse reveals specific distribution. This distribution presents seven maxima. According to these maxima the appropriate zones on Ramachandran map have been defined. Specificity of certain maxima is as follows: zone (code) C represents R-helical zone on Ramachandran ma. Zone E – β-structure. Z-zone – the right part of large β-structural area on Ramachandran map. Zone G – L-handed helix.
Differentiation between E and F zone bases on map representing R and V-angle. Zone E is represented by highest R – radius of curvature. F zone represents structures of lower radius of curvature. The conformations of F code are always accompanying the β-structural as terminating fragment where the gradual decrease of radius introduces bends or loops terminating the β-structural linear propagation.
Radius of curvature is calculated for pentapeptides of common Phi Psi angles following the step-wise covering the Ramachandran map with the 5 deg step. The pentapeptide for each conformation is commonly oriented: averaged positions of C atoms and O atoms (C=O group) determines the Z-axis. The projection of Cα atoms on XY plane allows calculation of radius of curvature. The V-angle expresses the angle between XY plane and plane of peptide bond for each point on Ramachandran map. Difference between i-th residue and i+1-th expresses the change of angles between two sequential peptide bond planes. It is easy to explain taking two secondary structures: α-helix and β-structure. In α-helix the V-angle is equal 0 deg while in β-structure this angle V=180 deg. Radius of curvature in helix is low, while in β-structure is almost theoretically infinitely large. All other structures represents status between 0 < V < 180 and radius of curvature 0.8 < R < 8 (in Ln scale). The relation between LnR as dependent on V-angle for low energy area on Ramachandran map is shown Fig 3.A. Interpretation of this curve is representation of the optimal backbone conformations. The points taken from Fig.5A localised on Ramachandran map are shown on Fig.5B. The ellipse was selected as representing the optimal path for conformational changes as the way to link all secondary structural forms. This is why the set of conformations belonging to ellipse is treated as early stage assuming no presence of any form interactions. The early stage is treated as optimal from the point of view of backbone.
The late stage of folding introduces all forms of interactions applied to early stage searching for minimum of energy. Additionally the hydrophobic interaction (FOD model) is introduced to late stage step of folding.
This is why the application of ellipse path to amyloid transformation was applied for amyloid transformation shown in Fig. 13 and 14.
The structures of early stage shown in Fig. 11. to transthyretin in two forms (aggressive and resistant to amyloid transformation) reveal the differences. The aggressive form reveals larger participation of linear β-structural form. It may suggests better “closure” in respect to linear forms present in amyloids.
The explanation and origin of zones is given in Supplementary Materials. The zones A-G are also shown in newly added Figure 4.C.
How is the set of examples shown in figure 7 obtained? The details are not precise enough to be sure of the methodology?
The new legend for Figure 7 – as expected -explains the main idea shown in this figure. The chain fragments are the same as in Figure 5. The intention of this Figure is visualisation of mutual orientation of H-bonds expressed by arrows. The black-white arrows visualise anti-parallel orientation of atoms building H-bonds generated by β-structural fragments. The red arrows represent the orientation of H-bonds based on single residue with R-α-helical or L-α-helical conformation which introduces the locally parallel orientation of H-bonds. The source of planar structure of the polypeptide chain is explained in this way. The combination of β-structural conformations with individual residue of R-α-helical or L-α-helical incorporated into β-structural fragments. Additionally the β-structural conformation to keep the linear construction of the chain (and in consequence planar) shall be constructed by the Phi, Psi angles satisfying the conditions Psi = -Phi. This equation does not expresses any changes of Phi, Psi angles. It expresses the mutual relation of dihedral angles producing linear form of chain. The visual expression of this observation is the localisation of Phi, Psi angles in amyloids. Almost perfect example is shown on Ramachandran map for Alpha-synuclein amyloid – 6PEO – see the map below.
As can be seen only few amino acids represent the conformation different than Psi = - Phi.
The paper discussing the alpha-synuclein amyloids is currently in preparation for publication. This tendency can be seen in all amyloid structures independently on the protein under consideration.
Figure 8 the red should be the orange. - CORRECTED
Do all the points move between R and L, or is it just an impression?
The positions of Phi, Psi angles calculated on the basis of structures available in PDB – 1DVQ – native, 6SDZ – amyloid form of transthyretin.
When you talk about the condition Psi = - Phi
Does this mean that 70 becomes -70, or just that the positive values become negative?
No – Relation Psi = -Phi expresses the characteristics of majority of conformation present in amyloid. The straight line (as it is present in amyloids – for almost all Beta-structural fragments) amino acids represent the Phi, Psi angles satisfying the relation Psi = -Phi. It means that on may find AA with Phi, Psi angles for example Phi = -100, Psi = 100. It is well seen on the Ramachandran maps – see for example map for 6PEO – amyloid Alpha-synuclein – This is the perfect example for this thesis.
Appropriate explanation added to the text. The explanation of this problem – as we assume – is shown on the figure above.
When you see the figure 13, it is normal to wonder if the resistant form and the aggressive form are not the same. Is there really a structural difference?
Figure 13 (currently 12) shows the possible hypothetical path of structural changes starting with native (A) and finishing in amyloid (I) structural forms using path proposed in this paper – following the ellipse path as shown in Figure 13.B. Step-wise structural changes shown in A are following the Phi, Psi angles changes shown in B.
Appropriate explanation added in text.
I didn't understand the discussion about complexes and dimers. Are we sure of anything?
This discussion is obvious taking the FOD model as the base for complex and amyloids formation.
The complexes (for example homodimers) demonstrate the local exposure of hydrophobicity on the surface. These two areas (discordant in respect to FOD model) avoiding the contact with water tend to create complex using these areas as interface construction avoiding the exposure of hydrophobicity exposure toward water. The generation of complexes does not require significant structural reorganisation – example – structures of different forms of Fab fragments of IgG. The monomers, individual domains, different complexes of these proteins are available in PDB. Meanwhile the structure of amyloid of V domain of this protein (6HUD) requires significant structural reorganisation. This protein is also object of our analysis based on FOD and early stage model of folding similarly to the discussion given in the paper under consideration now.
It is suggested that the amyloid can not be treated as IV-order structure. The best term is “associate”.
Section 2.5 is surely the most relevant and expected. The FOD approach seems to be particularly appropriate for understanding this type of biological property.
The explanation given above – we think – is appropriate for this point of revision.
Section 2.6 is a bit more difficult and especially the changes in Phi and PSi values are not clear. Is this just a follow-up of the proposed L2 and G2 lines?
It follows the ellipse path. The ellipse path is expressed by equation, while the lines G and L do not. This is why we are able to generate the Phie and Psie very precisely.
One shall point out very precisely – this model is of hypothesis form.
In figure 14 the question arises as to the number of residues on the ellipse. Is it significant or very punctual?
The positions of Phi, Psi angles shown on this Figure are selected in step-wise procedure. The native Phi, Psi angles are in different distance versus ellipse and Psi=-Phi. The procedure is :
- Approach the ellipse in step-wise procedure
- Reach the position observed in amyloid
- If the final conformation is of form Psi=-Phi the next steps approach this position.
The 3-points procedure (as given here) is just proposal for computer simulation of amyloid transformation. The simulation starts with native structure. The Phi, Psi angles get changed in simulation according to given 3-points procedure as long as the planar structure of the chain is reached.
The part on Abeta and well stretched obligatory in view of the subject but because of the length could go into additional material.
The Abeta amyloids got moved to Supplementary Materials.
The discussion is a bit disappointing. One would like to have a more critical view of the approaches used and the results proposed. Also, more thought should be given to the importance of the sequence in the process; is it taken into account in a fine enough way so that mutants with drastic Clear impacts can really be taken into account.
We did our best to present the applicability of the presented model. All red fragments are those newly implemented.
The material and method section is too short. One would like to have more intermediate details on the different calculations made.
The description of model delivering the ellipse path got extended.
So this paper is very interesting, but particularly long and it is sometimes difficult to know what messages each section brings. It would be good to reduce it a bit and to highlight more relevant results to help the reader to read it properly.
Many thanks for this opinion. We shall do our best to make our model comprehensible. We hope that the correction introduced made the paper easier for reading.

Round 2
Reviewer 2 Report
Many thanks for the quality of the answers to my many questions, I now have a complete picture of the work. The paper can be published as is.